# The Epidemiology of Chickenpox in England, 2016–2022: An Observational Study Using General Practitioner Consultations

**DOI:** 10.3390/v15112163

**Published:** 2023-10-27

**Authors:** Megan Bardsley, Paul Loveridge, Natalia G. Bednarska, Sue Smith, Roger A. Morbey, Gayatri Amirthalingam, William H. Elson, Chris Bates, Simon de Lusignan, Daniel Todkill, Alex J. Elliot

**Affiliations:** 1Real-Time Syndromic Surveillance Team, Field Services, Health Protection Operations, UK Health Security Agency, Birmingham B2 4BH, UK; megan.bardsley@ukhsa.gov.uk (M.B.); paul.loveridge@ukhsa.gov.uk (P.L.); natalia.bednarska@ukhsa.gov.uk (N.G.B.); sue.smith@ukhsa.gov.uk (S.S.); roger.morbey@ukhsa.gov.uk (R.A.M.); dan.todkill@ukhsa.gov.uk (D.T.); 2Immunisation and Vaccine Preventable Diseases Division, UK Health Security Agency, London NW9 5EQ, UK; gayatri.amirthalingam@ukhsa.gov.uk; 3Nuffield Department of Primary Care Health Sciences, University of Oxford, Oxford OX2 6ED, UK; william.elson@phc.ox.ac.uk (W.H.E.); simon.delusignan@phc.ox.ac.uk (S.d.L.); 4TPP SystmOne, Leeds LS18 5PX, UK; chris.bates@tpp-uk.com

**Keywords:** varicella, varicella-zoster, chickenpox, general practitioner, primary care, vaccination, syndromic surveillance

## Abstract

Chickenpox is a common childhood disease caused by varicella-zoster virus (VZV). VZV vaccination is not part of the UK childhood immunisation programme, but its potential inclusion is regularly assessed. It is therefore important to understand the ongoing burden of VZV in the community to inform vaccine policy decisions. General practitioner (GP) chickenpox consultations were studied from 1 September 2016 to 9 December 2022. Over the study period, the mean weekly chickenpox consultation rate per 100,000 population in England was 3.4, with a regular peak occurring between weeks 13 and 15. Overall, rates decreased over time, from a mean weekly rate of 5.5 in 2017 to 4.2 in 2019. The highest mean weekly rates were among children aged 1–4 years. There was no typical epidemic peak during the COVID-19 pandemic, but in 2022, rates were proportionally higher among children aged < 1 year old compared to pre-pandemic years. Chickenpox GP consultation rates decreased in England, continuing a longer-term decline in the community. The COVID-19 pandemic impacted rates, likely caused by the introduction of non-pharmaceutical interventions to prevent SARS-CoV-2 transmission. The lasting impact of the interruption of typical disease transmission remains to be seen, but it is important to monitor the chickenpox burden to inform decisions on vaccine programmes.

## 1. Introduction

The varicella-zoster virus (VZV) causes Varicella disease, commonly known as chickenpox, which is an extremely common infection in childhood and usually results in lifetime immunity. Symptoms include a characteristic itchy rash with small, fluid-filled blisters that eventually scab over. Although the disease is typically self-limiting, severe complications can develop, particularly in pregnant women, infants, and the immunocompromised [1]. In 2014, the World Health Organisation (WHO) estimated that over 4 million hospitalisations and over 4000 deaths occurred globally each year due to complications associated with VZV infection [2]. In England, it is estimated that the majority of the hospital burden due to VZV occurs in the immunocompetent population and is therefore potentially vaccine-preventable [3].

VZV is highly contagious and is spread person-to-person through direct contact with the blisters, saliva, or mucus of an infected person. VZV can also be transmitted through the air via coughing and sneezing [4]. There are effective, live-attenuated vaccines to prevent chickenpox disease, and 36 countries (2019) have introduced universal vaccination against VZV (including the United States, Canada, Australia, and some European countries, including Spain, Germany, and Italy) [5]. However, the United Kingdom (UK) and many other countries have not implemented a universal programme, choosing instead to adopt a selective programme of vaccination offered to specific high-risk groups only, based on cost-effectiveness considerations [6]. Modelling predicts a decline in chickenpox that could lead to a consequent increase in zoster incidence in older age groups and subsequent increase in the disease complication rate [5,7].

Understanding the burden of VZV in the UK is challenging. Chickenpox is not currently a notifiable infectious disease, nor is it likely to result in all cases presenting to healthcare. Laboratory confirmation of cases is rarely sought, as diagnosis can often reliably be made on clinical grounds [8]. The diagnosis of chickenpox by general practitioners (GPs) in the primary care healthcare setting remains the only source of routinely available chickenpox surveillance data, which can provide insight into the overall community burden of disease and trends. In the UK, GP primary care services are the main route for patients to access the National Health Service (NHS). As clinical diagnosis of chickenpox in the UK is usually established based on the presenting symptoms and medical history, consultation data from GPs can be used as a form of syndromic surveillance, i.e., monitoring certain clinical indicators based on signs and symptoms, rather than relying on laboratory confirmation. Similar to other communicable diseases in England, it is likely that patterns of chickenpox transmission have been significantly impacted by the interventions imposed during the COVID-19 pandemic [9,10].

Here, we used routinely collected GP consultation data to report on recent changes in chickenpox epidemiology in England. We describe trends in GP consultations for chickenpox disease between 2016 and 2022 and examine differences by age and sex. Ultimately, this information is important for informing decision-making around vaccination policies.

## 2. Materials and Methods

### 2.1. Study Design and Population

This was a retrospective, observational descriptive analysis of routinely collected daily in-hours GP consultations for chickenpox disease across a network of GP practices in England. The study population was all persons who presented to GPs participating in the UK Health Security Agency (UKHSA) GP in-hours surveillance system between 1 September 2016 and 9 December 2022 (inclusive).

### 2.2. Case Definition

A case of chickenpox was defined as a GP consultation episode where the GP assigned either a Read or SNOMED-CT (Systematised Nomenclature of Medicine Clinical Terms) code (the classification systems currently used in UK practice) consistent with chickenpox based on the presenting symptoms and history as part of the routine clinical management of the patient (Appendix A).

### 2.3. Data Sources

The UKHSA was established in 2021 and is an executive agency of the UK Government, protecting the population from the impact of infectious diseases; chemical, biological, radiological, and nuclear incidents; and other health threats [11,12]. The remit of the UKHSA covers mostly the population of England, with other public health agencies (Public Health Scotland, Public Health Wales, and the Public Health Agency (Northern Ireland)) operating in the other UK nations.

The UKHSA coordinates a programme of real-time syndromic surveillance, based on a suite of national syndromic surveillance systems that monitor healthcare data from across the NHS in England. Syndromic surveillance can be defined as the process of collecting, analysing, and interpreting health-related data to provide an early warning of health threats requiring public health action [13]. This ‘all-hazard threat’ syndromic surveillance programme monitors daily emergency department attendances, calls to a telehealth telephone and online health service (NHS 111), GP in-hours and GP out-of-hours consultations, and ambulance dispatch call-outs.

The UKHSA GP in-hours surveillance system uses two sources of GP consultation data, which were used for the study: GP consultations provided by the Oxford-Royal College of General Practitioners Clinical Informatics Digital Hub (ORCHID), which (over the study period provided) had a coverage of 11.6 million patients in England, and The Phoenix Partnership (TPP), which provided coverage of a further 6.3 million patients. Both the TPP and ORCHID comprise the UKHSA GP in-hours surveillance system [14]. TPP and ORCHID data are routinely collated to give an overall a network of 1,932 practices in England with a coverage of 17.9 million patients. In England, general practice is a registration-based system with patients only registered with a single practice, enabling an accurate denominator for this study. The UKHSA GP in-hours surveillance system covers 31.5% of the England population (2021 England population estimate, 56.5 million [15]).

### 2.4. Statistical Analysis

Daily counts of GP consultations for chickenpox and the GP registered practice population were extracted for each day, age group, and sex from 1 September 2016 to 9 December 2022 (inclusive).

Mean weekly chickenpox consultation rates and 95% confidence intervals (CI) per 100,000 population across England were calculated using the weekly count of chickenpox consultations as the numerator and weekly GP registered population as the denominator. Weeks began on a Monday, and public holidays and weekends were removed to reflect most GP practice services being closed outside of usual working hours.

Time series graphs were used to visualise the secular trends and seasonality of the weekly national consultation rates for chickenpox, overall and stratified by age group and sex. The mean annual rate of consultations and range of weekly rates of consultations per year were calculated, and the peak week for chickenpox consultations was identified in each year. Consultation rates and rate ratios (RR) were calculated by age group and the sex of the patient. This was performed separately on data from before the COVID-19 pandemic (defined as consultations made before the week commencing 9 March 2021, when the first announcement of physical distancing was made by the UK Government [16]) and on data from during the COVID-19 pandemic (defined as consultations made from the week commencing 9 March 2021 onward [9]) to assess any changes in age and sex breakdown since the introduction of non-pharmaceutical interventions (NPIs) aimed at preventing COVID-19 transmission.

## 3. Results

A total of 198,692 GP in-hours consultations for chickenpox were reported to the GP in-hours system during working days between 1 September 2016 and 9 December 2022, with an overall mean weekly rate of 3.39 consultations per 100,000 GP registered patients (range, 0.23 to 9.87 per 100,000). From 2017 to 2019, peaks in consultation rates were observed at a similar time each year, between weeks 13 and 15 (Figure 1). Consultation rates reduced during the Christmas periods, due to there being fewer working days when GPs were open during this time. Over the study period, there was a decreasing long-term trend; the consultation rate decreased from a weekly mean of 5.52 per 100,000 in 2017 to 4.18 per 100,000 in 2019 (Figure 1, Table 1).

Following the introduction of the first COVID-19 restrictions in week 11 (early March) 2020, the chickenpox consultation rate dropped immediately, and there was no obvious typical epidemic peak throughout 2020 or 2021 (Figure 1). Rates started to increase at the start of June 2021; however, they did not increase past a weekly mean of 2.06 per 100,000 until 2022. The peak in 2022 occurred later (in week 21), and was of smaller magnitude (6.69 per 100,000), compared to pre-pandemic years (Figure 1, Table 1).

There was a difference in the chickenpox consultation rate across age groups. From 2017 to 2019, children aged 1–4 years had the highest mean weekly rate (41.90 per 100,000), followed by infants < 1 year of age (31.49 per 100,000) (Figure 2, Table 2). The relative GP consultation rate appeared to shift towards the youngest age group in the 2022 peak; children aged 1–4 years had a consultation rate 16% higher than infants aged < 1 year in the period during COVID-19, compared to a 33% higher rate during pre-pandemic years (Figure 3, Table 2).

Consultation rates were equal between males and females, and this did not change during the COVID-19 pandemic period (overall rate ratio of 0.99 for females compared to males, Figure 4, Table 2).

## 4. Discussion

This study provides a follow-up to previous UK-based VZV burden studies to show trends in community disease caused by VZV in England using routinely available chickenpox GP consultation data. As chickenpox is not currently a notifiable disease, and samples are rarely sent for diagnostic testing, the reliable and systematic reporting of chickenpox data is limited and relies on GP consultation data as the primary source of intelligence. Here, we demonstrate that chickenpox disease typically peaks during the start of the calendar year, during weeks 13–15, and that GP consultation rates have been falling since 2017. In the UK, GP consultations for chickenpox had been declining over a long period and, therefore, our results confirm this trend is continuing [17,18]. It is important to note, however, the reasons for this decreasing secular trend may be complex and reflect changes in healthcare utilisation by patients, a true decline in disease burden, or a combination of both (and other factors). Now, compared to previous decades, there may be a perception that chickenpox is not particularly severe for most children, and parents may decide not to seek any health advice at all and manage simply at home. Furthermore, over the last two decades, patients have increasingly had more options for seeking non-GP services, including pharmacies, emergency departments, telephone/online health advice services, and the Internet [19].

Across the study period, the highest incidence was observed among children aged 1–4 years, followed by infants aged < 1 year old. However, we show that the epidemiology of chickenpox in England has changed considerably since 2020, likely due to NPIs aimed at preventing the transmission of COVID-19 [20]. There was no typical epidemic peak observed in 2020 or 2021, and the peak in 2022 was delayed compared to previous years. The age profile of the disease has also shifted ‘post-pandemic’, with incidence rates among the < 1 and 1–4-years age groups more similar to each other compared to pre-pandemic years. A shift in the age-specific incidence of chickenpox has been previously reported. Using GP consultation rates, Ross and Fleming (2000) reported a downward shift in age-specific chickenpox incidence between 1983 and 1998; incidence doubled in children aged 0–4 years and halved in children aged 5–14 years [21]. They hypothesised that the shift had resulted from historical ‘cultural’ changes in childcare provision linked to an increasing trend for young infants and pre-school children (aged 0–4 years) to be placed into nurseries and pre-school facilities.

The sudden fall in chickenpox activity and change in seasonality from early March 2020 coincided with the introduction of NPIs in England. NPIs were primarily introduced at a national level (and then subsequently at regional levels) as part of the national management of the COVID-19 pandemic in England. During the first stringent ‘lock down’ periods, schools and nurseries were largely shut for several months, with the vast majority of children schooling from home. Access to healthcare services also changed; for example, there was a decrease in face-to-face consultations with GPs, while telephone consultations increased [22,23]. Other changes in healthcare seeking behaviour were also reported, including large decreases in emergency department attendances [24], suggesting that patients likely changed their behaviour due to concerns about exposure to COVID-19 [25]. It is probable, therefore, that the initial reduction in chickenpox activity monitored through national surveillance data was first driven by a reduction in healthcare seeking behaviour during more stringent lock down periods. The ongoing reduction in chickenpox activity could then have been propagated by a genuine reduction in VZV incidence in the community due to interrupted transmission chains (with schools and nurseries remaining closed).

A further observation in our study was a shift in the age profile of chickenpox consultations during 2022. Pre-pandemic, chickenpox activity was notably higher in children aged 1–4 years compared to infants aged < 1 year (RR = 1.33), likely reflecting social mixing patterns in this age group in nursery and pre-school settings. During 2022, rates in the 1–4-years age group decreased proportionally more than infants aged < 1 year, resulting in more similar rates between the age groups (RR = 1.14). This is likely due to a combination of complex reasons linked to the introduction of NPIs and social restrictions during that pandemic. The majority of nursery, pre-school, and school settings were closed, limiting opportunities for transmission outside of the family setting. The impact of NPIs on the circulation of other pathogens such as respiratory syncytial virus (RSV) during the immediate post-pandemic era has also been reported [9,26]. Another contributing factor could be behavioural changes, where parents of older children may not have attended their GP whilst those of younger children may have continued to seek healthcare, thus shifting relative age group proportions. Anecdotally, during 2022, UKHSA responded to an unusually high number of chickenpox outbreaks in educational settings (Amirthalingam G, UKHSA, personal communication), which again demonstrates the unusual character of post-pandemic chickenpox activity. An increase in school-based outbreaks with no corresponding increase in GP consultations may reflect the fact that once an outbreak has been confirmed in a school, local health protection procedures would dictate that the school and affected parents are informed of the outbreak. This might therefore reduce the likelihood of those parents (with children presenting with early signs and symptoms of chickenpox) consulting a GP if they are already aware of chickenpox cases in their child’s school.

Overall, the continuing decreasing secular trend of chickenpox reported here continues a series of historical studies reporting similar findings in the UK [17,18]. With respect to informing decisions on VZV vaccination, the continuing reduction in the incidence of chickenpox in younger cohorts of the population will increase susceptibility to chickenpox infection in the future. Severity, complications, and hospitalisations associated with chickenpox increase with age, and therefore, there are implications with respect to the cost effectiveness of chickenpox vaccination in the UK [1,19]. However, a more likely impact of the shift from pre-school to school-aged children is an increased workload for local public health and health protection teams in dealing with more complex chickenpox outbreaks in the school setting.

For respiratory viruses such as RSV and influenza, England saw high inter-seasonal surges of infection in summer 2021, corresponding with the lifting of COVID-19 NPIs. However, this effect was not observed for chickenpox. COVID-19 NPIs were eased after the natural seasonal peak of chickenpox, which should have happened in April 2021, and therefore, we postulate that the seasonal, environmental, and social conditions at that point were not optimal for chickenpox to spread. It is also possible that continued heightened public knowledge of infection control and continued social distancing contributed to lessening the spread of disease.

A strength of our study is the use of a routine, systematic, and ongoing surveillance system that monitors GP consultations for chickenpox. This system routinely monitors a consistent and representative population of approximately 18 million patients across England [27]. However, there were several limitations to our study. The outcome we monitored was clinically diagnosed chickenpox without virological confirmation. This may represent a classification bias (the inclusion of false positives and exclusion of true positives) and, therefore, we may under- or over-estimate chickenpox burden if there was misdiagnosis by the GP. However, chickenpox has a characteristic clinical presentation, and misdiagnosis by GPs is therefore unlikely. We did not include virological data in this study as laboratory diagnosis of VZV is rarely performed routinely in England. The main aim of this study was to utilise the routinely available GP in-hours data. However, we do recognise that other sources of data informing on the chickenpox burden in England, including hospitalisations, are available [1]. A further limitation of using GP in-hours data was that it was difficult to interpret chickenpox trends during the COVID-19 pandemic. In England, GP practices reduced face-to-face consultations, particularly during periods of COVID-19 social restrictions (NPIs) [22,23]. It is therefore hard to tell, for the pandemic years, how much of the reduction in chickenpox rates was due to actual reduction in circulating virus versus an artefact of GP practices reducing in-person consultations (although telephone consultations would still have been used and captured in our data). Furthermore, a diagnosis of chickenpox is highly visual, so parents may have been more likely to search the Internet for making a diagnosis at home, not needing to consult with a healthcare professional. A GP may also have been less likely to diagnose chickenpox in the absence of a visual examination of the child. This means that the data we present are likely an underestimate of the true burden of disease in the community, but these data allow for accurate trends and descriptive epidemiology of presentations caused by the disease. For future studies, seroprevalence data could provide additional information on the incidence of infection in the population. For studies of this type, it is often an advantage to analyse relevant vaccine update data in parallel with health data to infer any likely associations between the two variables. Universal vaccination against VZV is not part of the UK routine childhood immunisation programme, and, therefore, vaccine uptake data are not routinely monitored or available. It is not possible to analyse uptake in those populations which are selectively targeted (e.g., immunocompromised patients). Furthermore, the market for private VZV vaccination in England is relatively small and, therefore, unlikely to influence the secular chickenpox trends described through the GP consultation data.

This study has important findings relevant to the continued assessment of the VZV vaccination programme in the UK. Currently, the chickenpox vaccine is not part of the routine childhood vaccination schedule and is recommended only for close contacts of vulnerable patients, such as those who are immunocompromised. It is important, however, to continually monitor the incidence of chickenpox to identify any changes in burden that might indicate a need to consider or update the vaccine programme [28]. Having established and effective surveillance systems in place is critical to enable horizon-scanning for changes in chickenpox epidemiology, thereby generating the surveillance data to support the evidence base that might influence those future decisions.

## Figures and Tables

**Figure 1 viruses-15-02163-f001:**
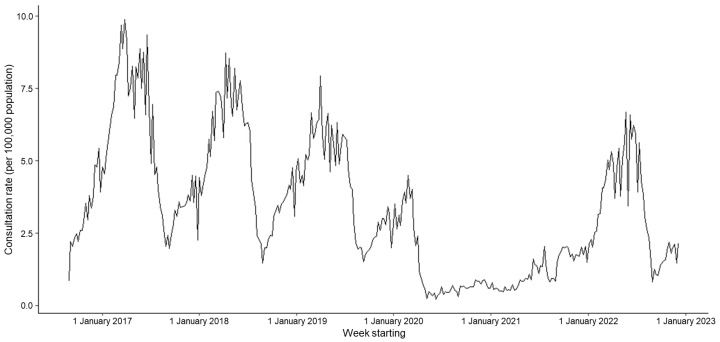
Weekly consultation rate for chickenpox per 100,000 registered population from participating general practitioners (GPs) contributing to the GP in-hours syndromic surveillance system, England, 1 September 2016 to 9 December 2022.

**Figure 2 viruses-15-02163-f002:**
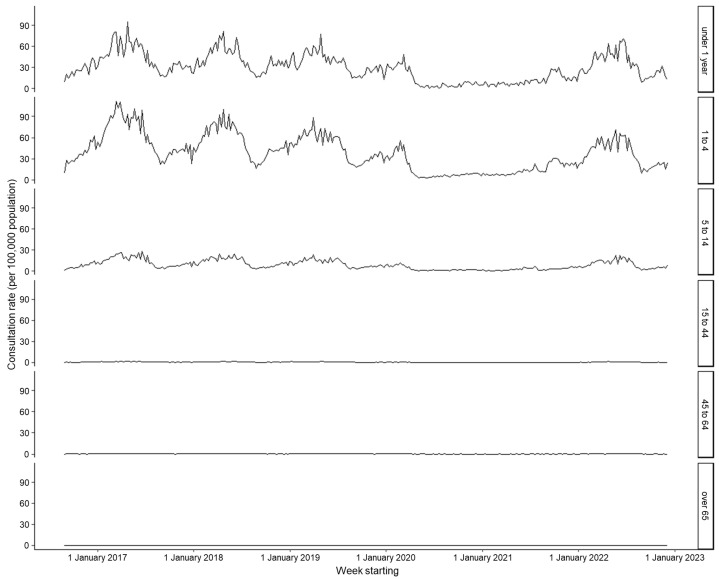
Weekly consultation rate for chickenpox per 100,000 registered population from participating general practitioners (GPs) contributing to the GP in-hours (GPIH) syndromic surveillance system, England, by age group, 1 September 2016 to 9 December 2022.

**Figure 3 viruses-15-02163-f003:**
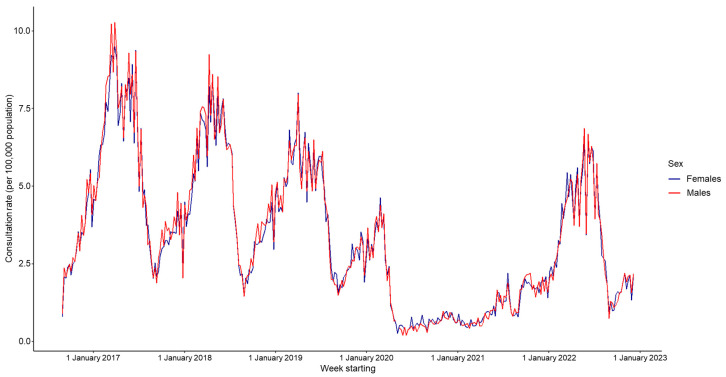
Weekly consultation rate for chickenpox per 100,000 registered population from participating general practitioners (GPs) contributing to the GP in-hours (GPIH) syndromic surveillance system, England, by sex, 1 September 2016 to 9 December 2022.

**Figure 4 viruses-15-02163-f004:**
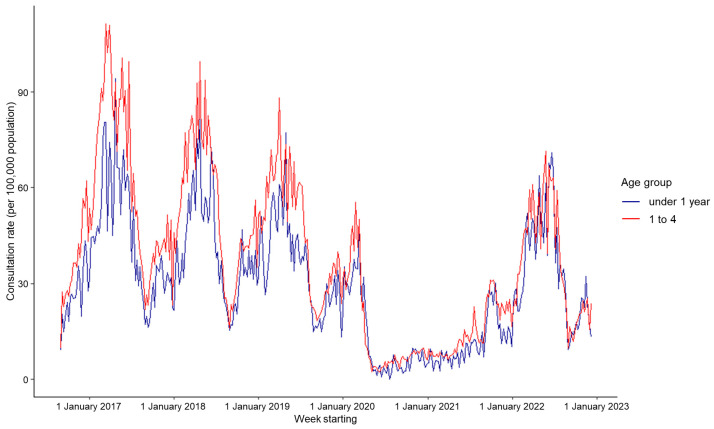
Weekly consultation rate for chickenpox per 100,000 registered population from participating general practitioners (GPs) contributing to the GP in-hours (GPIH) syndromic surveillance system, England, for infants aged < 1 year and children aged 1–4 years old, 1 September 2016 to 9 December 2022.

**Table 1 viruses-15-02163-t001:** Mean weekly rate of chickenpox consultations reported to the general practitioner (GP) in-hours syndromic surveillance system per 100,000 GP registered patient population by year, 95% confidence interval (CI), and activity range, 2017–2022.

Year	Mean Weekly Rate ^1^	Lower 95% CI	Upper 95% CI	Activity Range (Rate)		Peak Week Number
				Minimum	Maximum (Peak Week)	
2017	5.52	4.86	6.18	1.97	9.87	13
2018	4.88	4.33	5.42	1.46	8.73	15
2019	4.18	3.70	4.66	1.53	7.93	14
2020	1.28	0.93	1.62	0.23	4.49	-
2021	1.20	1.05	1.35	0.48	2.06	-
2022 ^2^	3.36	2.88	3.84	0.84	6.69	21

^1^ Rate per 100,000 registered patient population; ^2^ 2022 includes data up to 9 December 2022.

**Table 2 viruses-15-02163-t002:** Mean weekly consultation rates and rate ratios for chickenpox, from participating general practitioners (GPs) contributing to the GP in-hours syndromic surveillance system, by age group and sex, for the pre-COVID-19 and COVID-19 periods.

	Whole Study Period (1 September 2016 to 9 December 2022)		Pre COVID-19 (1 September 2016 to 8 March 2020)		Post COVID-19 (9 March 2020 to 9 December 2022)	
Characteristic	Rate per 100,000	RR ^1^	Rate per 100,000	RR	Rate per 100,000	RR
Age						
<1	29.53	-	31.49	-	24.41	-
1 to 4	38.01	1.29	41.90	1.33	27.86	1.14
5 to 14	8.96	0.30	9.73	0.31	6.97	0.29
15 to 44	0.67	0.02	0.73	0.02	0.50	0.02
45 to 64	0.19	0.01	0.21	0.01	0.13	0.01
Over 65	0.09	0.00	0.10	0.00	0.06	0.00
Sex						
Male	3.43	-	3.81	-	2.44	-
Female	3.38	0.99	3.74	1.02	2.45	1.00

^1^ Rate ratio.

## Data Availability

Applications for requests to access relevant anonymised data included in this study should be submitted to the UKHSA Office for Data Release. Available at: https://www.gov.uk/government/publications/accessing-ukhsa-protected-data (accessed 24 October 2023).

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
