# Peer review of "The Epidemiology of Chickenpox in England, 2016–2022: An Observational Study Using General Practitioner Consultations"

_viruses, 2023, doi:10.3390/v15112163_

Round 1

Reviewer 1 Report

Comments and Suggestions for Authors

This is a well-written study Bardsley, Elliot and colleagues. I think it adds valuable information to the field especially given the lock-down periods the study encompasses. I have a few suggestions and edits for consideration.

1) Can the authors include data on the percent uptake trends of the vaccine among eligible individuals? Is this trend increasing and thus giving rise to the decrease in cases over the years? 

2) The last sentence of the paper mentions the importance of continued monitoring of incidences incase changes in the burden occur. This reviewer agrees. One point that could also be mentioned is the risk of the age of kids when chickenpox manifests. For instance, voluntary vaccination (which disproportionately favors wealthier families) has the risk of protecting a subset of children. The partial herd immunity will decrease the potential exposure rates to the unprotected (less wealthy and statistically less healthy), likely pushing the age of chickenpox manifestation to older kids. The older the individual when they contract varicella, the more dangerous is becomes. Thus, a voluntary vaccination program for a virus that gets progressively more dangerous as you increase in age could have a disproportionate burden on lower income families within the next decade. 

Discussion lines 244-246: The authors state that RSV and influenza are more infectious than varicella. Varicella has a much higher R0 value than both RSV and influenza, therefore, I do not think this statement is accurate. 

Reviewer 2 Report

Comments and Suggestions for Authors

In this manuscript entitled "The epidemiology of chickenpox in England 2016-2022: an observational study using general practitioner consultations", Elliot and colleagues provide epidemiological data of varicella incidence in England for more than 6 years. This includes the data during COVID-19 pandemic, and this would adds more value on this manuscript. While authors do not discuss how this data would affect the vaccine program in England in which  routine vaccination against varicella has not been incorporated, these accumulation of epidemiological evidence might become important not only for medical society but also for social welfare. Overall the manuscript is important to publish and the reviewer hopes continuous this type of research, especially regarding varicella epidemiological change during "real" post-COVID era.

There are minor issues but please check carefully.

1. Are the legends for Figure 3 and 4 OK? For the reviewer, these are mixed up. The statements about Figure 3 and 4 (lines 141-161) also seems mixed up.

2. Line 167; Could you say "almost a decade" for 6-year data?

3. Lines 244-246; This statement is not acceptable and seems wrong. VZV is highly contagious as authors state in the introduction and eventually VZV's basic reproduction number (R0) is the second or third highest among any known human specific viruses. Furthermore VZV mainly transmits via airborne route especially during varicella.

Reviewer 3 Report

Comments and Suggestions for Authors

The paper by Bardsley et al presents and analyzes GP consultation rates for varicella in the UK over a 9-year period, specifically showing pre-pandemic versus post-pandemic GP consultation rates. The study is interesting and shows us in detail the history of GP consultation rates in the population covered by GPs participating in the UK Health Safety Agency's (UKHSA) GP Surveillance System. However, after reading the paper, I have some questions and doubts that need to be clarified, in order to make the article even more meaningful. Specific comments are shown in the attached document.

Round 2

Reviewer 3 Report

Comments and Suggestions for Authors

The authors responded satisfactorily to most of the comments. It is a bit disappointing that the authors chose not to expand the scope of the work to other data sources, but limited themselves to routinely available chickenpox data. Certainly, this study provided a starting point for further studies on the burden of chickenpox in the English population. An additional paragraph in lines 95-109 of the methods section provides more information about UKHSA but is too detailed and does not answer the main question: what proportion of the population of England is covered by UKHSA real-time syndromic surveillance? I suggest to shorten this paragraph and include data about the size of the registered GP population on which this study results are based. What proportion of the population is it in relation to the total population of England?

Author Response

Please see attached response. 
